# A Machine Learning-Based Correlation Analysis between Driver Behaviour and Vital Signs: Approach and Case Study

**DOI:** 10.3390/s23177387

**Published:** 2023-08-24

**Authors:** Walaa Othman, Batol Hamoud, Alexey Kashevnik, Nikolay Shilov, Ammar Ali

**Affiliations:** 1Saint Petersburg Federal Research Center of the Russian Academy of Sciences (SPC RAS), 199178 St. Petersburg, Russia; walaa_othman@itmo.ru (W.O.); bkhamud@itmo.ru (B.H.); nick@iias.spb.su (N.S.); 2Institute of Mathematics and Information Technologies, Perozavodsk State University (PetrSU), 185035 Petrozavodsk, Russia; 3Information Technology and Programming Faculty, ITMO University, 191002 St. Petersburg, Russia; ammarali32@itmo.ru

**Keywords:** correlation analysis, vital signs, machine learning, driving behaviour, driver maneuvers, external events

## Abstract

Driving behaviour analysis has drawn much attention in recent years due to the dramatic increase in the number of traffic accidents and casualties, and based on many studies, there is a relationship between the driving environment or behaviour and the driver’s state. To the best of our knowledge, these studies mostly investigate relationships between one vital sign and the driving circumstances either inside or outside the cabin. Hence, our paper provides an analysis of the correlation between the driver state (vital signs, eye state, and head pose) and both the vehicle maneuver actions (caused by the driver) and external events (carried out by other vehicles or pedestrians), including the proximity to other vehicles. Our methodology employs several models developed in our previous work to estimate respiratory rate, heart rate, blood pressure, oxygen saturation, head pose, eye state from in-cabin videos, and the distance to the nearest vehicle from out-cabin videos. Additionally, new models have been developed using Convolutional Neural Network (CNN) and Bidirectional Long Short-Term Memory (BiLSTM) to classify the external events from out-cabin videos, as well as a Decision Tree classifier to detect the driver’s maneuver using accelerometer and gyroscope sensor data. The dataset used includes synchronized in-cabin/out-cabin videos and sensor data, allowing for the estimation of the driver state, proximity to other vehicles and detection of external events, and driver maneuvers. Therefore, the correlation matrix was calculated between all variables to be analysed. The results indicate that there is a weak correlation connecting both the maneuver action and the overtaking external event on one side and the heart rate and the blood pressure (systolic and diastolic) on the other side. In addition, the findings suggest a correlation between the yaw angle of the head and the overtaking event and a negative correlation between the systolic blood pressure and the distance to the nearest vehicle. Our findings align with our initial hypotheses, particularly concerning the impact of performing a maneuver or experiencing a cautious event, such as overtaking, on heart rate and blood pressure due to the agitation and tension resulting from such events. These results can be the key to implementing a sophisticated safety system aimed at maintaining the driver’s stable state when aggressive external events or maneuvers occur.

## 1. Introduction

Given the fundamental role that driving plays in modern society, the importance of ensuring safety on the roads cannot be overstated [1]. Despite extensive efforts to reduce the number of fatalities resulting from car accidents, the magnitude of the problem remains deeply worrying [2]. In order to combat this trend, governments and stakeholders must prioritize the implementation of measures designed to reduce the risks of accidents and associated injuries and fatalities [3]. One critical aspect that deserves close consideration is the role that external factors and deliberate decisions to alter driving styles can play in impacting drivers’ emotional and physical states, potentially leading to greater risk [4]. Proactive measures must be taken to ensure that drivers maintain safe driving practices and remain mindful of the ways in which external factors can affect their driving behaviour and other vital signs. To this end, many studies have been undertaken to observe and investigate the influence of external factors and driving behaviour on vital signs such as heart rate and blood pressure [2,5,6]. However, these investigations have often been limited by their narrow focus on events either inside or outside the vehicle, as well as by the use of cumbersome devices to observe a single vital sign in most cases.

Accordingly, our paper outlines a comprehensive investigation into how the factors related to external events, the driver’s maneuvers, and the distance from the nearest vehicle impact the drivers’ state—including vital signs such as heart rate, blood pressure, oxygen saturation, and respiratory rate—as well as their eye state, indicated by whether the eyes are open or closed, and head pose, specified by yaw, pitch, and roll angles.

In order to detect the driver’s maneuvers, the smartphone sensors were leveraged, specifically the gyroscope and accelerometer, and a decision tree classifier was used to determine whether the driver was performing a maneuver or not. An external events classification model was also trained, incorporating Convolutional Neural Networks (CNN) [7] and Bidirectional Long Short-Term Memory (BiLSTM) [8], to classify the actions of other drivers in the vicinity of the vehicle. Furthermore, several contact-less approaches were employed to obtain the driver’s vital signs, eye state, and head pose along with the proximity to other vehicles using deep learning-based models from our previous work that require videos taken by a smartphone camera to process the external view or the face and other parts of the body to estimate the aforementioned variables. The contributions of this paper can be summarized as follows:Implementing an end-to-end study that utilizes a combination of sophisticated techniques to integrate and analyse disparate data streams to provide new insights into driver behaviour, external events, and potential hazards on the road.Employing contact-less technologies to bring forth a more holistic view of the correlation between a driver’s state (including vital signs, eye state, and head pose) and external events, driver maneuvers, and other critical factors.Generating actionable insights that can be used to promote safer driving behaviours and reduce accidents on the road.

This rest of the paper is as follows: Section 2 represents our motivation to run this study. Section 3 contains a review of the existing methods to detect the driver’s maneuvers or behaviour. Section 4 includes a description of the presented approach to perform correlation analysis of the relationship between the external events, vehicle maneuvers and driver’s state. Section 5 includes a description of the datasets used for external events and maneuvers classification followed by the results of these classification procedures and the correlation analysis between the driver’s parameters and both maneuvers and external events. Section 6 outlines the study and the obtained results, including the future plans and limitations.

## 2. Motivation

Many papers worked on studying the impact of the driver behaviour and external factors outside the vehicle on the driver’s vital signs such as blood pressure and heart rate. One of these studies [2] performed analysis of the relationship between driving aggressiveness and heart rate, and observed that the heart rate of participants engaged in aggressive behaviour was on average 2.5% to 3% higher compared to those who remained calm. Another paper [5] showed by experiments that the heart rate increases when associating driving with other activities or cognitive workload that requires much attention, observation, and mental effort. In addition, the authors of [6] analysed in their work the impact of exposure to traffic congestion on the blood pressure of the driver, and they noticed that longer exposure time was associated with higher systolic and diastolic blood pressure levels since traffic congestion might trigger an inconvenient atmosphere which can be associated with anger, stress, or frustration that cause an increase in the blood pressure levels. These findings suggest that a driver’s behaviour and the external factors during driving can have a notable impact on physiological indicators. Therefore, these studies and findings were the motivation to analyse the relationship among the vital signs and the maneuvers performed by the driver and the external events performed by other drivers or pedestrians. This idea came after implementing contact-less approaches to estimate the vital signs (heart rate, blood pressure, oxygen saturation, and respiratory rate) [9,10,11,12] and the head pose of the subject [13] by processing his/her facial video, which provided for an opportunity to employ an end-to-end study of the relationship between the driver parameters and his/her maneuvers along with the external events happening outside the vehicle. Our dataset provides synchronized videos for inside and outside the cabin collected from several drivers with data obtained from gyroscope and accelerometer sensors in the driver’s smartphone. Therefore, training new models to detect the maneuvers and external events was the only step left to begin our analysis and investigation.

## 3. Related Work

Smartphone sensors can be utilized for various purposes, such as keeping track of human physical activity [14], identifying transportation modes [15], and categorizing driving actions [16]. These sensors can also be employed to recognize specific risky situations during a trip, such as driving under the influence or aggressive driving. However, many studies worked on the tasks of event detection and identifying the driver’s current behaviour with phone sensors by extracting specific features from sensor data and using machine learning models to create the final classifier [17]. Therefore, several studies that implemented machine learning (including deep learning) methods to achieve a classifier or detector for the driver maneuvers or behaviour during his/her trip are discussed in this section.

The authors of [18] utilized the accelerometer sensor of a smartphone installed inside a vehicle to detect the driver’s maneuver and assess the kinematic condition of the vehicle. Their classification method integrated four vehicle state classes (stopped, driving, parking, and parked) and utilized three classifiers: Random Forests (RF), Support Vector Machines (SVM), and Fuzzy Rule-based Classifiers (FRC).

The authors of the paper [19] used smartphone and OBD-II data to detect driver behaviour. To gather data regarding throttle, speed, and revolutions per minute (RPM), an OBD-II adaptor was employed, while acceleration and gravity data were collected by a smartphone securely fixed inside the car. This method included two steps to identify risky driving behaviour. By applying a time window to the signals that were gathered and a recurrence plotting approach to the windowed data, time-dependent input signals are first transformed into spatially dependent images. The image is then classified into five categories of driving behaviour in the second step using a CNN, including normal, aggressive, distracted, drowsy, and drunk driving.

The authors of [16] developed a simulation using a car kinematic model to train an SVM classifier. They then tested the trained model on driving data obtained by using smartphone sensors. To overcome the issue of unpredictable phone orientations within a vehicle, the suggested method utilized Principal Component Analysis (PCA) on gyroscope data to calibrate the gyroscope rotation matrix. The analysed driving maneuvers comprised stopping, acceleration, deceleration, and left and right turns.

Other researchers [20] employed a cloud-based approach to classify various driver actions, including overtaking, stopping, stopping at traffic lights, and maintaining a safe distance. They processed the standard signals that are usually measured in a car, such as the speed, the engine revolutions (RPM), the angle of the steering wheel, the position of pedals, and others, without additional intelligent sensors. The classification process utilizing synthetic data was carried out using a fuzzy rule-based method.

The authors of [21] employed an interesting approach for detecting maneuvers using GPS, accelerometers, and gyroscopes. They employed a range of neural network models, including Feed Forward Neural Network (FFNN), Convolutional Neural Networks (CNNs), LSTM-Recurrent Neural Networks (LSTM-RNNs), and a stacked ensemble of the best models (Stacked Neural Network SNN) for classification. To improve the accuracy of their models, they also performed several preprocessing steps, including normalizing the phone orientation, using the Kalman filter to remove noise, and transforming the data to make them suitable for neural network training.

Table 1 includes an outline of the above-mentioned approaches including the implemented algorithms and methods, as well as the data type used to detect the preformed maneuver or the driver’s behaviour.

To summarize, it is noticeable that the aforementioned articles managed to achieve maneuvers classification using data collected by smartphone-embedded sensors, however, they had some drawbacks such as the requirement of other data (e.g., OBD-II data) and high computational cost (using filters and deep learning algorithms). Therefore, this paper introduces a method that provides a low cost and time efficient approach by using only the data readings from gyroscope and accelerometer sensors embedded in a smartphone to detect the maneuvers performed by the driver using ensemble learning algorithms, since it has proved itself in the field of achieving classification with good and high accuracy especially when dealing with small or unbalanced datasets. This maneuvers classifier is used for further experiments to study the effect of the occurrence of maneuvers on the driver’s vital signs as explained further in Section 4.5.

## 4. The Proposed Approach

This section includes the details of our approach consisting of three main methods (in-cabin video analysis, out-cabin video analysis, and maneuver classification) which leads us to the core of our work to investigate the relationship between the external events and the driver’s state, study the correlation between his/her vital signs and the performed maneuvers, as well as analyse the correspondence between driver’s vital signs and the distance from the closest vehicle.

### 4.1. Methods for In-Cabin Video Analysis

This subsection briefly describes our previously developed methods used for analysing the driver parameters including vital signs (blood pressure, respiratory rate, heart rate, and oxygen saturation [9,10,11,12]) as well as the driver head pose [13].

For estimating the respiratory rate, a detection of the chest keypoint at each frame of the driver’s video is first performed, then using an optical flow-based algorithm, the displacement between the frames is calculated, after that, post-processing techniques on the obtained chest movement signal are applied, such as filtering and denoising, and finally the number of real peaks in this signal is counted [11].

For estimating the heart rate, the face and facial landmarks using 3DDFA_V2 framework [22] are extracted, then a pretrained model based on the 3D variant of EfficientNet-B1 is used, followed by a simple classifier consisting of a linear layer [9].

As for estimating the blood pressure, 3DDFA_V2 is used again to detect the face and extract the left and the right cheeks, each of which was fed into pretrained model (CNN) followed by a long short-term memory (LSTM) model to make final estimations for systolic and diastolic blood pressure using fully connected layers [10].

Regarding the estimation of the oxygen saturation, a similar to the blood pressure prediction approach is used by replacing the LSTM with XGBoost Regressor, since ensemble learning is a better option due to the biased nature of the oxygen saturation values to be concentrated between 90–100% [12].

The estimation of the head pose (Euler angles: roll, pitch, and yaw angles) is achieved using the method described in [13]. To estimate the head pose, the face first is detected using YOLO tiny. After detecting the face, a 3D face reconstruction is used to fit facial landmarks even if they are not visible to the camera, then the facial landmarks are detected and the Euler angles are calculated by finding the transition and rotation between the landmarks in successive frames.

The eye state is estimated using trained model [23]. The model takes the face detected by FaceBoxes [24] as an input and outputs whether the eye is opened or closed.

### 4.2. Method for Out-Cabin Video Analysis

This subsection concisely describes our proposed monocular depth estimation method [25,26]. We have collected data from more than 10,000 videos recorded by our Drive Safely system [23] over a period of more than five years, capturing various scenes, lighting, and weather conditions. Four different state-of-the-art methods were used to pseudo-label these data with approximate depth maps. Then, we designed a lightweight neural network architecture based on the EfficientNet-B0 feature extractor with nested UNet decoder using a complex loss function based on mean absolute error, cosine similarity loss, Sobel filter, and virtual normal loss. It allowed us to achieve competitive results to the state-of-the-art methods for publicly available datasets with 15x faster performance compared to the AdaBins model on the RTX 3090 GPU card. The model was trained on our data using pseudo-labeling and estimated the relative distance between the camera and other vehicles from the predicted relative depth.

### 4.3. Maneuver Classification Method Based on Sensor Data

This subsection describes the proposed method to classify the maneuvers that the driver performs while driving based on the data obtained from the gyroscope and accelerometer sensors embedded in any smartphone, since the estimated maneuvers will be used to study their relationship with human’s vital signs (respiratory rate, heart rate, blood pressure, and oxygen saturation).

For training, Section 5.1.2 dataset was used after being divided into 217 samples for training and 53 samples for testing. Given the size of the training set, deep neural network-based learning methods were not used due to the high probability of having overfitted models. Therefore, several methods of training based on ensemble learning were applied. The following methods were used: XGBoost classifier, decision trees, random forest, SCV, and SGDclassifiers. Weights have been added to low frequent classes (aggressive left lane change and aggressive right lane change) to make sure that the dataset is balance in order to achieve acceptable accuracy despite the small size of the dataset. These methods were fed by 24 features (mean, maximum, minimum values, and standard deviation) of the accelerometer and gyroscope readings on the three axes (ax, ay, az, gx, gy, and gz) of each consecutive three seconds, and for each training process, we tried different parameters such as the number of the estimators (num of estimators), maximum depth (max depth), number of features considered (max features), and others. The table includes the best parameters for each method when achieving the highest accuracy (Equation (Equation 1)) using this specific method. However, the best method among the others was chosen to proceed with our experiments.
(1)Accuracy=(TP+TN)/(TP+TN+FP+FN),
where *TP* presents True positive, *TN* donates True negative, while *FP* and *FN* are False positive and False negative, respectively.

### 4.4. Investigating the Relationship between the External Events and the Driver State

This subsection describes our developed model for external events classification in addition to the overall scheme of calculating the correlation between the external events and the driver state.

#### 4.4.1. Correlation Analysis between External Events and the Driver Parameters

The main goal of this section is to investigate if there is any relation between the external events detected by our developed method explained in Section 4.4.2 and the driver state (eye state) and the driver head pose (Euler angles: roll, pitch, yaw) [13] from one side and the driver vital signs from the other side. Figure 1 shows the overall scheme to calculate the correlation matrix.

As shown in the figure, our proposed model was trained on the Meteor dataset [27]. Synchronized videos for inside and outside the cabin collected from several drivers [28] were used to check the relation between the driver state and the detected event. For the inside videos, we used the eye state detection model and the head pose estimation model [13] to estimate the driver state, as well as the respiratory rate, heart rate, blood pressure, and oxygen saturation estimation models to estimate the driver vital signs. For the outside videos, the pretrained model was used to predict outside events such as overtaking and changing lanes. Then, the information from the models was merged to calculate the correlation matrix based on Pearson’s correlation coefficient (Equation (Equation 2)).
(2)$$r=\frac{\sum_{i=1}^{n}\left(x_{i}-\bar{x}\right)\left(y_{i}-\bar{y}\right)}{\sqrt{\sum_{i=1}^{n}{\left(x_{i}-\bar{x}\right)}^{2}}\sqrt{\sum_{i=1}^{n}{\left(y_{i}-\bar{y}\right)}^{2}}}$$
where *r* is the correlation coefficient, $x_{i}$ and $y_{i}$ present values of the x-variable and y-variable, respectively, $\bar{x}$ shows the mean of the values of the x-variable, and $\bar{y}$ is the mean of the values of the y-variable.

#### 4.4.2. External Events Classification Model

This subsection covers the proposed method for detecting external events occurring in front of the vehicle such as overtaking, cutting, changing lane, and yielding. For detecting the external events, we considered the problem as event classification with an artificial “no event” class. We proposed to use the Resnet50d feature extractor followed by Bidirectional LSTM (BiLSTM) to capture both spatial and temporal features, then three fully connected layers were used as a classifier.

Resnet50d is a modification of the ResNet architecture that utilizes an average pooling tweak for down-sampling so no information is ignored. The motivation is that in the unmodified ResNet, the 1 × 1 convolution for the down-sampling block ignores 3/4 of the input feature maps. The Resnet50d feature extractor consists of five layers followed by a global feature pooling layer. The first layer consists of three convolution blocks (two convolution blocks + batch normalization + rectified linear unit (ReLU)) followed by max pooling. The second layer consists of three bottlenecks, and each bottleneck in turn consists of three convolution blocks. The third layer consists of four bottlenecks, while the fourth layer consists of six bottlenecks, and the last layer consists of three bottlenecks.

The input of the model is 10 successive frames after applying resizing to (512 × 512) and normalization, and the output is either no event or one of the following events: overtaking, yielding, cutting, or lane changing. Figure 2 shows the proposed method for outside event classification. For training the neural network cross entropy (Equation (Equation 3)) was used as the loss function, Adam optimizer to update the weights, and the ReduceLROnPlateau as the scheduler to reduce the initial learning rate (initially chosen to be 1 × 10^−3^) when the accuracy stopped improving.
(3)$$CrossEntropy=-\sum_{i=1}^{C}t_{i}\log\left(s_{i}\right)$$
where $t_{i}$ is the ground truth, $s_{i}$ is the score for each class *i*, while *i* is the class index, and *C* indicates the number of classes.

### 4.5. Correlation Analysis between Vital Signs and Maneuvers

The workflow of estimating the correlation between the vital signs of the driver and the maneuvers of the vehicle is shown in Figure 3. First of all, video recordings of the driver from the cab of the vehicle are processed by heart rate, blood pressure, and oxygen saturation models in order to obtain vital signs. The event detection model described in Section 4.3 requires gyroscope and accelerometer data for the previous three seconds, so to obtain the necessary values, vital signs are also averaged over the last three seconds. Further, the data obtained from all models are stored in a CSV file to check the correlation (Equation (Equation 2)) between heart rate, blood pressure (systolic and diastolic), and blood oxygen saturation with the maneuver. It should be noted that no conditions were set regarding the speed of the car, since the heart rate, blood pressure, and oxygen saturation models require recording only the face area, which is not related to other circumstances, such as speed, acceleration, etc. In addition, a high or a moderate speed is necessary to perform the maneuvers, so the speed range included in the dataset used varies as needed, without affecting the values of heart rate, blood pressure, and oxygen saturation.

The workflow shown in Figure 3 contains three averaging units that process the results of the vital signs models. The purpose of using these averaging units is to synchronize the vital signs with the detected maneuvers, because sequences of gyroscope and accelerometer data based on the previous three seconds to detect the maneuver are used. So each unit averages consecutive values that describe three consecutive seconds of the obtained vital signs. Thus, averaging allows to obtain synchronized data consisting of the heart rate, the systolic and the diastolic blood pressure, the oxygen saturation of the blood, and the presence of maneuvers that are stored in a CSV (Comma-separated values) file. This file is used later to check for the correlation between vital signs and the occurrence of a maneuver. It should be mentioned that the output of the maneuvers detection model is the class of the maneuver, but it was converted into 1 to indicate that there is a maneuver being performed and 0 otherwise, because the occurrence of the maneuver provides more significance than the class and it affects the vital signs whatever the maneuver class is.

### 4.6. Correlation Analysis between Vital Signs and Distance

The workflow shown in Figure 4 describes the analysis of the correlation (Equation (Equation 2)) between the respiratory rate, the heart rate, the blood pressure (systolic and diastolic), and the oxygen saturation with the distance between the current car and the nearest car in front of it.

Previously developed models were used to obtain heart rate, blood pressure, oxygen saturation, and respiratory rate, as well as the model for estimating the distance to the vehicle in front for each frame of the out-cabin video. Since the operations of these models are not synchronized, it is necessary to perform an additional averaging operation. Thus, the estimation of the distance to the forward vehicle calculated for each frame are additionally processed by the averaging unit to match the data of the heart rate, the blood pressure, and the oxygen saturation calculated for each second. The processing results are saved in a CSV file. This file is analysed for the presence of a correlation between the distance and these vital signs.

On the other hand, since the respiratory rate estimation model calculates the respiratory rate once per minute, it was proposed to apply averaging of the data stored in the above-mentioned file to obtain the values of the vital signs (heart rate, systolic blood pressure, diastolic blood pressure, and oxygen saturation) and the distance to the vehicle in front for each minute synchronized with the respiratory rate. The result is also stored in a CSV file for subsequent verification of the correlation between these features.

In addition to the models that calculate heart rate, blood pressure, and oxygen saturation, it can be noticed that the workflow mentioned in this subsection also includes a model to predict the respiratory rate. However, this model is capable of estimating the respiratory rate once per minute and requires a vehicle speed of no more than 3 km/h [11], so that the signal extracted from the video is not distorted or noisy. This model is used only in this analysis, due to the fact that the videos used in the analysis of the correlation between vital signs and maneuvers have high speed during most of the trip, and in general, the video recordings of the parts of the trips with low speed have short duration (less than 1 min), which makes the RR estimation inapplicable in the workflow mentioned in Section 4.5.

## 5. Experiments and Evaluation

This section contains the descriptions of the used datasets in order to accomplish external events and maneuvers classification, followed by the results obtained regarding these two tasks to analyse the relationship between the driver’s state and external events, performed maneuvers, and distance from the nearest vehicle.

### 5.1. Used Datasets

This section includes the characteristics of the used datasets to achieve the external events classification and detecting the maneuver which the driver carries out based on the gyroscope and accelerometer readings collected by smartphone-embedded sensors.

#### 5.1.1. Used Dataset for External Events Classification

We used the METEOR dataset [27], which consists of 1250 one-minute videos. The dataset contains frame-wise annotations for agents and their behaviours on the road in diverse traffic scenarios including rainy weather, nighttime driving, driving in rural areas with unmarked roads, and high-density traffic scenarios. It contains the annotations for the following events.

Overtaking: an agent overtakes another agent with sudden or aggressive movement.Yielding, Cutting: a pedestrian or slow-moving agent tries to cross the road in front of another agent. If the latter slows down or stops, letting them cross the road, then such behaviour is labeled as yielding. If not, and the former agent’s action was interrupted, then the behaviour is labeled as cutting.Lane change: An agent aggressively changes lanes on roads with clear lane markings.

The METEOR dataset was used to train the external events classification model while synchronized inside-outside videos from the DriverMVTdataset [28] were used to check the correlation.

#### 5.1.2. Used Dataset for Maneuver Classification

This subsection describes the dataset used to detect maneuvers carried out by the driver based on sensor readings. The dataset [29] provides a collection of smartphone sensor measurements for maneuvers occurring while driving a car. A smartphone app is used to record smartphone sensor data (accelerometer, linear acceleration, magnetometer, and gyroscope) while the driver performs certain maneuvers while driving. This dataset covers an experiment involving four car rides of approximately 13 min each and includes a total of 69 events, divided as follows in Table 2.

The ground truth CSV file contains the start and end times of the maneuvers, as well as the maneuver class. To process the dataset, the time series of accelerometer and gyroscope readings on the three axes (ax, ay, az, gx, gy, gz) were divided into short series of three seconds long, and then the average value, standard deviation, maximum value and minimum value for each series were calculated and these features were labeled with the class of maneuver that occurred in the last second of these three seconds, therefore, 24 features were obtained for each sample, and these features were the input of our classifier. However, since the dataset is relatively small, this process was applied only for the periods when the maneuvers occurred, as well as a few samples when nothing happens, labeled as “no maneuver” to distinguish it from a “non-aggressive maneuver”.

The resulting dataset has the following number of samples:44 samples for the “non-aggressive” maneuver.39 samples for the “aggressive turn-right” maneuver.58 samples for the event “no maneuver”.37 samples for the “aggressive turn-left” maneuver.10 samples for the “aggressive right lane change” maneuver.10 samples for the “aggressive left lane change” maneuver.28 samples for the “aggressive break” maneuver.44 samples for the “aggressive acceleration” maneuver.

### 5.2. External Events Classification Results

The METEOR dataset has been split into training dataset and testing dataset, with the most common split ratio being 80:20. In splitting, it was ensured that the training and testing datasets have the same distribution of the external events classes. The model achieved a testing accuracy of 92.56%. Figure 5 shows the confusion matrix of the tested dataset, and Table 3 shows the model accuracy per class.

From Table 3, the accuracy for all external factors exceeds 82% except cutting, which is the case when any slow vehicle or pedestrian is interrupted by another vehicle. The main reasons for such results are the small number of samples for this class in the training dataset, and the similarity between the cutting and the yielding events. One way to improve the results is to merge the cutting and the yielding events into one class (jaywalking). If the results only for testing without retraining the whole model were combined, the overall accuracy for this class would be equal to 84.27%.

### 5.3. Maneuvers Classification Results

This subsection includes the results of the ensemble learning methods to estimate the maneuvers performed by the driver based on the accelerometer and gyroscope readings. Table 4 includes a list of the ensemble learning methods used in our study with the best parameters that led to the highest accuracy.

When analyzing the results, it can be seen that the best accuracy of 77% was obtained by the decision tree classifier. Hence, decision tree was used for further experiments in our study to find the correlation among the vital signs and the maneuvers. Figure 6 shows the confusion matrix for determining the accuracy of the estimation of the decision tree on the test set.

By analyzing Figure 6, it can be concluded that the proposed model can correctly identify most classes, including those that do not have a large number of samples in the training dataset. This means that adding weights to these classes solved the problem of lack of data, and the constructed model can recognize these classes with good accuracy, which is important because in real scenarios the range of maneuvers that the model should recognize (including those presented in the training dataset in insufficient quantity) can be quite wide. In addition, a conclusion was made about the ability of ensemble learning to classify events in even rarely occurring classes (including in training samples), which proves the flexibility of ensemble learning when working with unbalanced datasets. It can also be concluded that the classification of the presence of a maneuver could be a very informative sign, since vital signs depend not on the class of the maneuver, but on its presence, and based on the confusion matrix presented above, it can be seen that the model is able to distinguish between the presence or absence of a maneuver.

### 5.4. Correlation Analysis between Outside Events and Driver Behaviour

Our proposed model was tested on 64 synchronized pairs of videos from our dataset [28] of an average length of 1 min; one pair captures the driver while the other is capturing the road. Both videos were processed and the head pose and the eye state as well as outside events were calculated. Figure 7 shows the correlation matrix between the outside events and the head pose and the eye state.

Figure 7 shows a weak correlation between the yaw angle of the head (the angle of the driver head when he/she turns the head to the left or to the right) and the overtaking event (when a vehicle overtakes the ego vehicle) with a correlation coefficient of 0.22. To be sure that the result was statistically significant, the *p*-value between the yaw and overtaking was calculated and it gave 0.0001. The *p*-value is used to test the null hypothesis (the null hypothesis of a test is always aimed to predict no effect or no relationship between variables). If the *p*-value is less than 0.05, which was achieved in our case, it means that the results are statistically significant, and the null hypothesis is false. Since the *p*-value in our case is 0.0001, it means that our results are statistically significant. To interpret the results, values shown in Table 5 and mentioned in paper [35] were used.

It can be also noticed that there is a very weak correlation between eye state and head angle. This can be explained by the fact that since the camera is installed in front of the driver, when he/she is moving the head to certain positions, the eyes may appear closed, while in fact they are open, which leads to some correlation between head tilt angles and eye condition.

Additionally, a lack of correlation between the driver’s eye condition or head angles and events such as yielding, cutting, or changing lanes can be noticed. This is due to the fact that these events take place entirely in front of the driver. Overtaking, on the other hand, happens when a vehicle first travels from the rear side and then changes lanes in front of the ego vehicle.

### 5.5. Correlation Analysis between External Events and Driver Vital Signs

For the 64 in-cabin videos from our dataset [28], the driver vital signs (heart rate, blood pressure, and oxygen saturation) were calculated to investigate the correlation with the external events. The correlation matrix between the external events and the driver vital signs was found. Figure 8 shows the correlation matrix between the outside events and the driver vital signs.

The obtained matrix shows that there is a moderate negative relationship between heart rate and blood pressure with correlation coefficients of −0.38 and −0.32 for systolic and diastolic blood pressure, respectively. A very weak correlation between heart rate and overtaking (correlation coefficient 0.14) can also be noted, as well as a negative weak correlation between blood pressure and overtaking (correlation coefficients −0.25 and −0.18 for systolic and diastolic blood pressure, respectively). Results are interesting, but it should be noted that these results require further study, as they were obtained only on the basis of 64 videos with an average length of 1 min for the same person, filmed on three different roads during periods of low traffic.

### 5.6. Correlation Results between the Maneuvers and the Vital Signs

For this task, we also used our dataset [28], which contains video recordings of the driver associated with gyroscope and accelerometer data. Based on the set, a correlation was calculated between heart rate, blood pressure (systolic and diastolic), oxygen saturation, and the vehicle maneuver. Two experiments were performed. The first experiment was focused to check whether there was a correlation between maneuvers and vital signs taken from the same video recording, so that the conditions of the trip were the same throughout the entire data. Figure 9 shows the resulting correlation matrix with the *p*-value of each correlation coefficient. The following abbreviations are used in the figure: hr: heart rate, sb: systolic blood pressure, db: diastolic blood pressure, and os: the degree of oxygen saturation of the blood.

It can be noted that there is a significant correlation between all vital signs and the presence of maneuvers. The correlation coefficients between blood pressure and the maneuvers have a medium negative value, which indicates a moderate relationship between these signs, and the *p*-value (<0.05) indicates a significant correlation that can be explained by human nature: concern and/or concentration causes a decrease in blood pressure values in general. In addition, there is a weak positive correlation between heart rate and maneuvers, which is also expected, since a person’s heart can beat faster when performing any action that requires attention.

In the second experiment, scenes taken from different trips recorded in-the-wild in the vehicle cabin were included. The speed changes based on outside situation around the vehicle and the driver performs maneuvers in a natural manner. Also, all possible maneuvers in the trip were included in our analysing set to get enough samples of maneuvers so the set is not biased, which enables us to obtain relatable results (46 and 283 samples with different maneuvers in the first experiment and the second one, respectively). The result obtained is shown in Figure 10, and it is close to the result of the first experiment, although it has smaller coefficient values, which can be explained by an increase in the sample size. However, the result shows that there is a significant positive correlation between heart rate and maneuvers, and a significant negative correlation between blood pressure (both systolic and diastolic) and maneuvers. These results are in line with expectations. However, the correlations between the maneuvers and the vital signs (hr, sb, and db) are in the same direction in both experiments with *p*-value < 0.05, which indicates that the results are valid regardless the conditions of the trip and emphasizes the significance of the findings regarding the influence of performing maneuvers during driving on the vital signs in general.

### 5.7. Correlation between the Distance Vehicle in Front and Vital Signs

For this experiment we used our dataset that includes video recordings of the trip conditions inside and outside the car. However, to analyse the correlation between vital signs and distance, two correlation matrices were calculated, the first of which describes the correlation between heart rate, systolic & diastolic blood pressure, oxygen saturation, and distance based on the values obtained every second. The first matrix is shown in Figure 11.

Based on the results presented in Figure 11, it can be concluded that systolic blood pressure shows a significant, but very weak correlation with the distance to the forward vehicle, which indicates a slight change in the values of systolic blood pressure when the distance between vehicles changes, that is, the distance can affect the driver state.

The correlation between vital signs, including respiratory rate and distance with a sampling rate equal to one value per minute is shown in Figure 12. Based on the figure, it cannot be concluded whether there is a significant correlation between distance and any vital sign, although the calculated coefficients are quite high, but the *p*-values are higher than 0.05. This can be explained by the relatively small number of samples, which can lead to an inaccurate assessment of the significance of the correlation.

### 5.8. Summary of the Correlation between the Vital Signs and Both: External Events and Driver’s Maneuver

Figure 13 represents a recap of the findings and observations concluded through our study from Section 5.5 and Figure 10 and Figure 11. It can be observed that the heart rate and the blood pressure (systolic and diastolic) are in general affected by performing maneuvers. This can be noticed by the positive correlation coefficient for the heart rate, which shows that performing a maneuver while driving would raise the rate of heart beats as a sign of being nervous, worried, or focused, and this explanation applies to the negative correlation coefficient between the maneuvers and both systolic & diastolic blood pressure. In addition, based on the results obtained, “overtaking” is the most critical action that has noticeable and observable impact on the vital signs based on the significant correlations obtained based on the presented matrix. This correlation is considered reasonable, regarding the fact that this action has sudden and aggressive scenarios more than the circumstances of the lane changing and yielding. However, the correlation among the distance and the vital signs (heart rate and blood pressure) in general was insignificant (except for systolic), hence, it can be assumed that the distance from the frontal vehicle does not have such a noteworthy or serious influence on the internal state of the driver.

## 6. Conclusions

This paper presents an end-to-end study of the relationship between the driver state (vital signs, eye state, and head pose) and both the vehicle maneuver actions as well as external events, including the proximity to other vehicles in synchronized in-cabin/out-cabin videos. To achieve this study, new models were presented to classify the external events from videos with a total accuracy of 92.56% on the Meteor dataset by using hybrid deep learning methods (CNN and BiLSTM). Additionally, the paper introduces an implementation of maneuvers performed by the driver detection based on accelerometer and gyroscope sensor data provided by smartphone-embedded sensors using several ensemble learning to get an accuracy of 77% obtained by a decision tree classifier. However, the accuracy can be improved by including more samples. By this point, an employment of our existing models from our previous work was only needed to estimate vital signs (heart rate, blood pressure, oxygen saturation, respiratory rate), eye state (open or closed), head pose (roll, pitch, yaw), and the distance to the nearest vehicle, hence, all these variables were obtained from our dataset that provides synchronized in-cabin/out-cabin videos with accelerometer and gyroscope sensors reading.

Our investigation of the correlation among these variables indicates that there is a positive correlation between the heart rate and the maneuver action with a correlation coefficient of 0.14, and the overtaking external event with a correlation coefficient of 0.14, which can be explained by the complex interactions between the sympathetic and parasympathetic branches of the autonomic nervous system. Therefore, when a person undergoes a maneuver or caution event, there is a shift in the autonomic nervous system balance, leading to altered HRV metrics. This study also demonstrates a negative correlation between blood pressure (systolic and diastolic) and both the overtaking and the maneuver-action along with a negative correlation between the systolic blood pressure and the distance to the nearest vehicle with a correlation coefficient of −0.13. These physiological responses in blood pressure can be interpreted as a manifestation of feelings such as nervousness, anxiety, or concentration. In addition, the outcomes suggest a correlation between the yaw angle of the head and the overtaking event with a correlation coefficient of 0.22, which seems to be reasonable since external events during driving, such as overtaking, may trigger involuntary head movements and rotations, affecting the yaw angle of the head due to the fact that head orientations are important factors in analyzing the driving environment to be aware of the hazards. Nonetheless, these results are in line with expectations, which spotlights the efficiency of our models used to implement this investigation and highlights the significance of our research and outcomes.

However, these results should be interpreted with caution and further study is required due to the fact that some of these results were obtained from 64 videos with an average length of one minute for the same person, captured in low-traffic periods. Another limitation of this study is the error of the used models. As mentioned in the paper, the maneuver actions, the external events, the the proximity to other vehicles, the vital signs, the driver’s eye state, and the head pose were estimated using pre-trained models with varying error ranges. These errors in estimation also affect the obtained results.

Avenues for future research include collecting more synchronized in-cabin/out-cabin videos from different drivers during traffic and on different roads and recalculating the correlation between the external events, distance from the nearest vehicle, and the driver state. In addition, the performance of the maneuvers detection model can be enhanced by collecting more samples to expand the dataset. However, these findings and knowledge can be integrated in the future into a driving monitoring system that would make a noteworthy influence on the existing driving safety systems by keeping the driver state under observation and taking into account both external and internal factors affecting drivers and passengers.

## Figures and Tables

**Figure 1 sensors-23-07387-f001:**
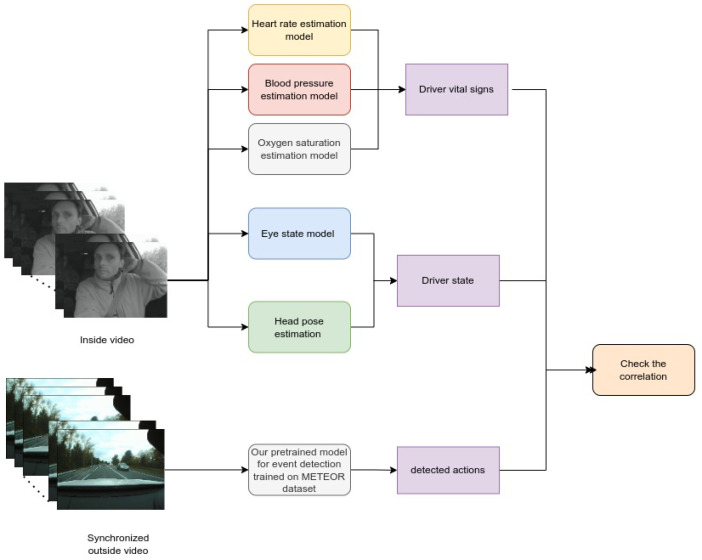
Proposed method for Correlation Analysis.

**Figure 2 sensors-23-07387-f002:**
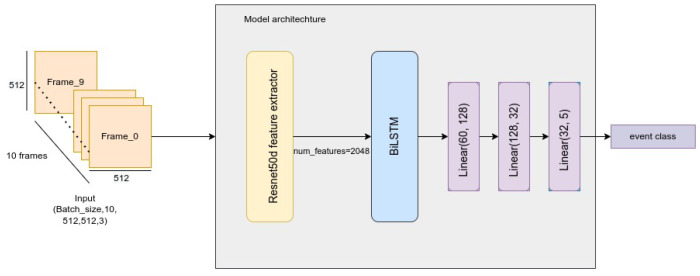
Proposed method for outside event classification.

**Figure 3 sensors-23-07387-f003:**
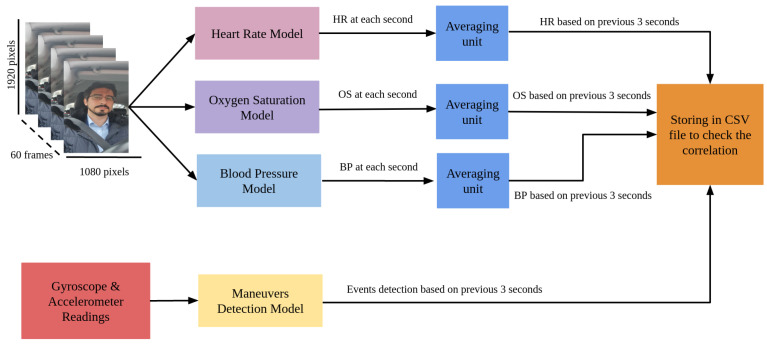
Workflow for assessing the correlation between vital signs and maneuvers.

**Figure 4 sensors-23-07387-f004:**
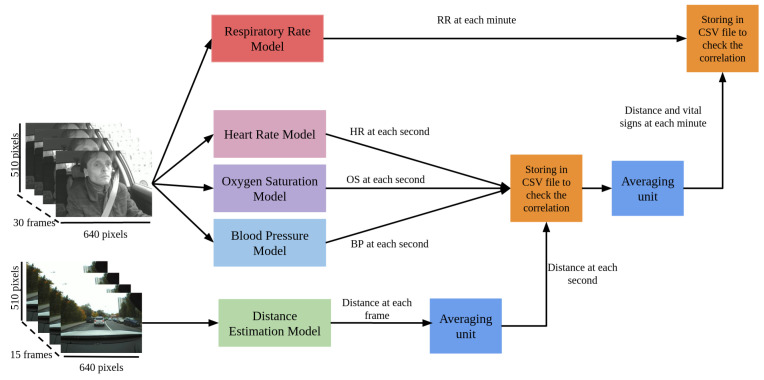
Workflow for correlation assessing between vital signs and distance to the vehicle in front.

**Figure 5 sensors-23-07387-f005:**
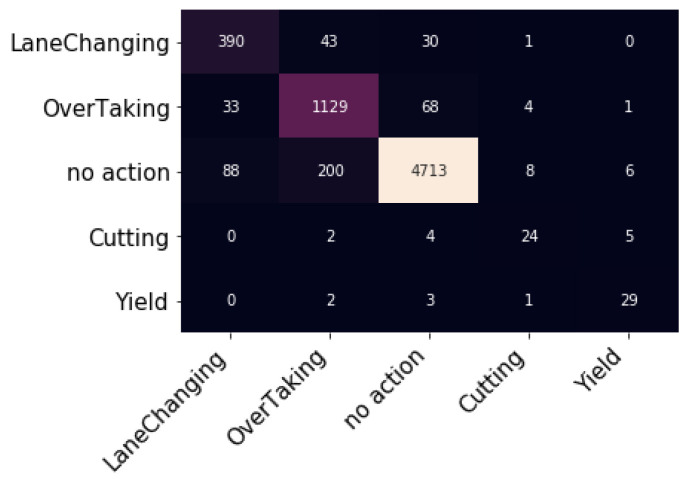
The confusion matrix of the tested dataset.

**Figure 6 sensors-23-07387-f006:**
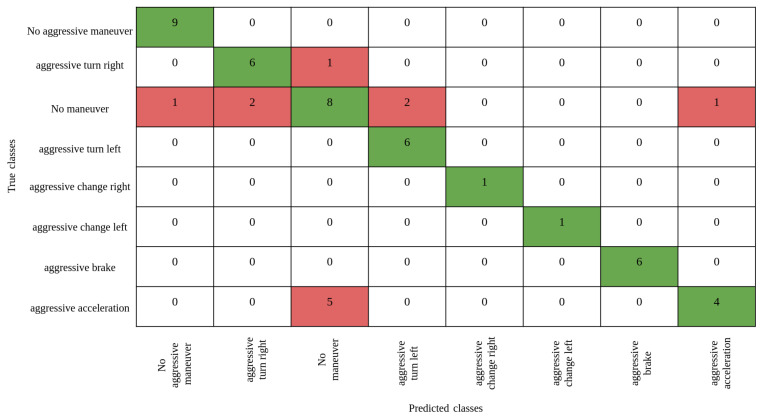
Confusion matrix for the decision tree classifier.

**Figure 7 sensors-23-07387-f007:**
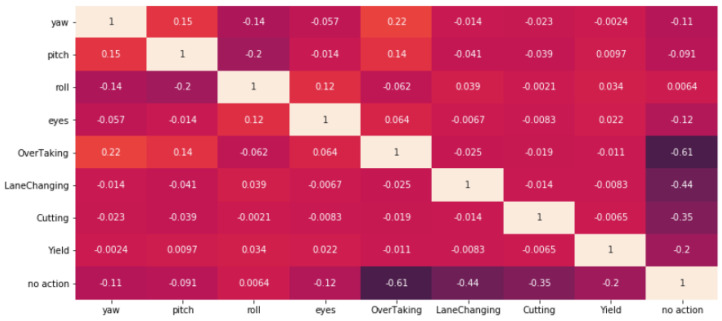
Correlation matrix between the outside events and head poses & eye state.

**Figure 8 sensors-23-07387-f008:**
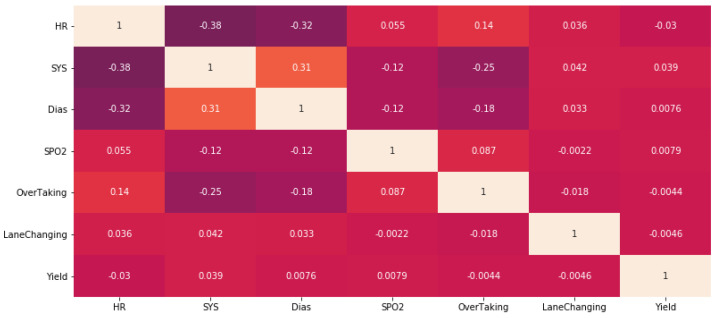
The correlation matrix between the outside events and the driver vital signs.

**Figure 9 sensors-23-07387-f009:**
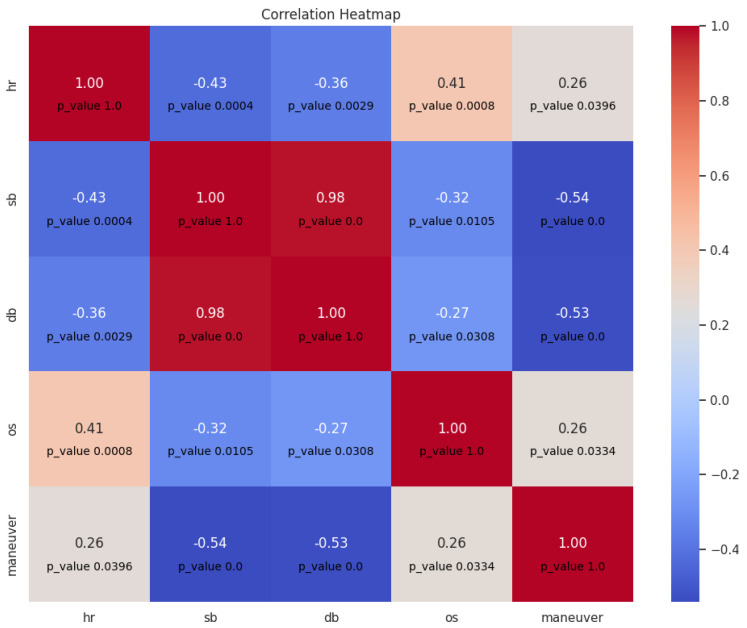
Correlation matrix between maneuvers and vital signs (Experiment 1).

**Figure 10 sensors-23-07387-f010:**
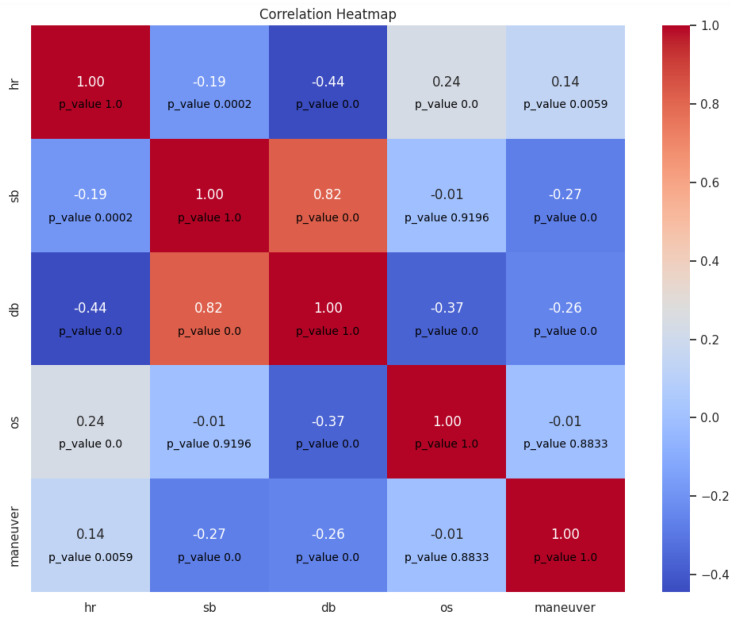
Correlation matrix between maneuvers and vital signs (Experiment 2).

**Figure 11 sensors-23-07387-f011:**
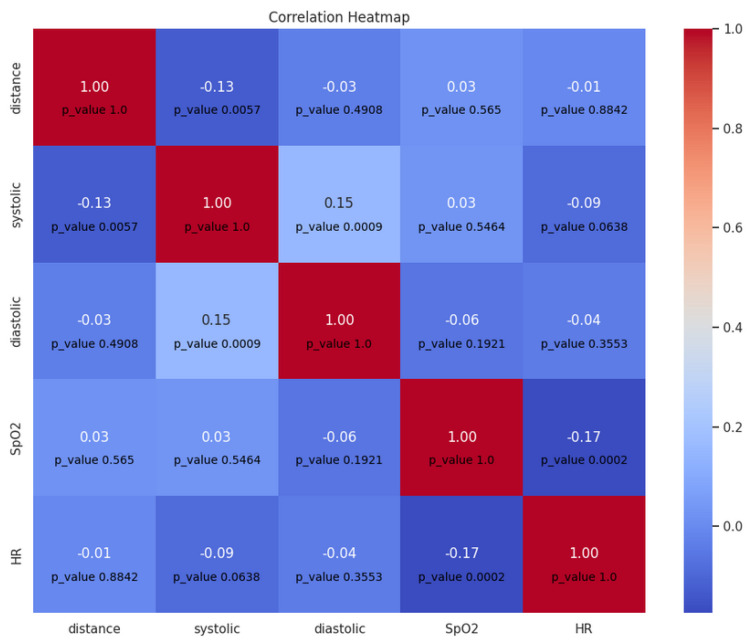
Correlation matrix between distance and vital signs per second.

**Figure 12 sensors-23-07387-f012:**
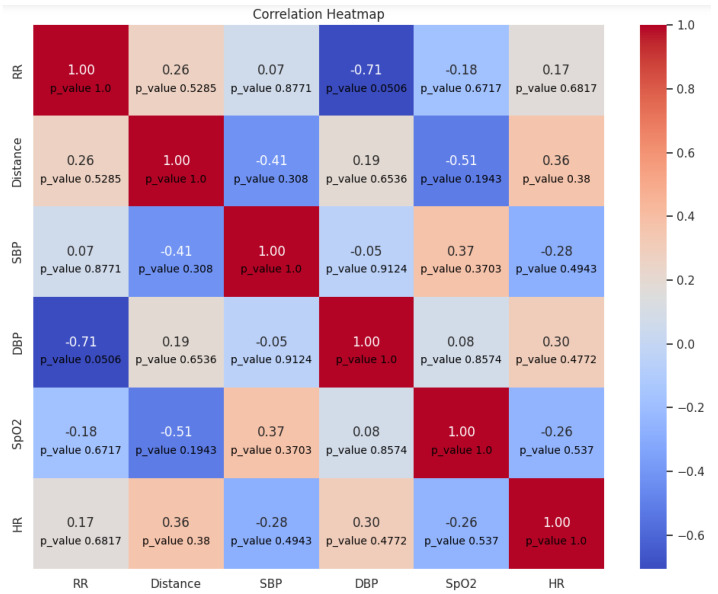
Correlation matrix between distance and vital signs per minute.

**Figure 13 sensors-23-07387-f013:**
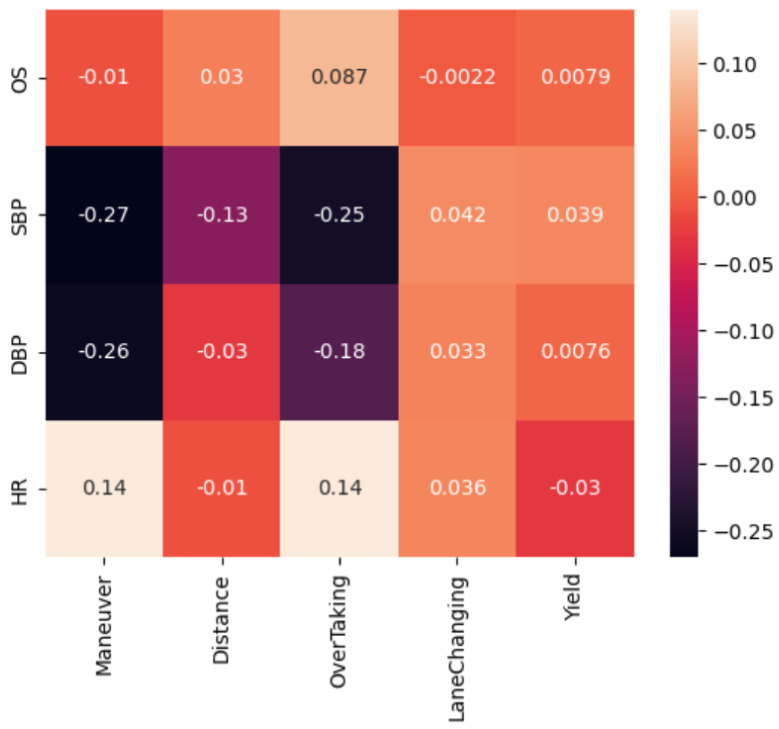
Final correlation matrix of the driver vital signs and vehicle behaviour.

**Table 1 sensors-23-07387-t001:** Summary of the existing approaches with used methods and data type.

Paper	Used Methods	Used Data
Cervantes-Villanueva et al. [18]	Random Forests (RF), Support Vector Machines (SVM), Fuzzy Rule-based Classifiers (FRC)	Accelerometer sensor data
Shahverdy et al. [19]	Convolutional Neural Networks (CNN) after converting time-dependent input signals into spatially dependent images	Throttle, speed, and revolutions per minute (RPM) using an OBD-II adaptor, as well as acceleration and gravity data were collected by a smartphone
Woo et al. [16]	Support Vector Machines (SVM), in addition to using Principal Component Analysis (PCA) for calibration purposes	Gyroscope sensor data
Andonovski et al. [20]	Fuzzy rule-based method	Speed, revolutions per minute (RPM), steering angle, pedal position, etc., without additional intelligent sensors
Ramah et al. [21]	Feed Forward Neural Network (FFNN), Convolutional Neural Networks (CNNs), LSTM-Recurrent Neural Networks (LSTM-RNNs), and Stacked Neural Network (SNN)	GPS, accelerometer, and gyroscope data

**Table 2 sensors-23-07387-t002:** Dataset characteristics.

Driving Maneuver Type	Number of Samples
Aggressive breaking	12
Aggressive acceleration	12
Aggressive left turn	11
Aggressive right turn	11
Aggressive left lane change	4
Aggressive right lane change	5
Non-aggressive event	14
Total	69

**Table 3 sensors-23-07387-t003:** The model accuracy per external event.

External Event	Accuracy (%)
Lane Changing	84.05
Over Taking	91.42
no external event (no action)	93.98
Cutting	68.57
Yield	82.86

**Table 4 sensors-23-07387-t004:** Results of the ensemble learning methods for the maneuvers classification.

Method	Hyper-Parameters	Accuracy
XGBoost [30]	max depth = 4, num of estimators = 10	39%
Decision Tree classifier [31]	criterion: ‘entropy’, max depth: 7, max features: 13	77%
Random forest [32]	criterion: ‘entropy’, max depth: 6, max features: 11	47%
Support Vector Classification (SVC) [33]	max depth = 2, kernel = ‘linear’	26%
Stochastic Gradient Descent (SGD) Classifier [34]	kernel = ‘huber’, penalty = l2	28%

**Table 5 sensors-23-07387-t005:** Interpretation of the correlation results.

Coefficient Interval	Correlation
0.00–0.19	Very Weak
0.20–0.39	Weak
0.40–0.59	Medium
0.60–0.79	Strong
0.80–1.00	Very Strong

## Data Availability

Not applicable.

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
