# Peer review of "A Machine Learning-Based Correlation Analysis between Driver Behaviour and Vital Signs: Approach and Case Study"

_sensors, 2023, doi:10.3390/s23177387_

Round 1

Reviewer 1 Report (Previous Reviewer 1)

The paper provides a research for Correlation Analysis Between Driver Behaviour and Vital Signs, however, the authors should consider the following comments for enhancing the content quality of the manuscript.

  1. The title has not clearly written, I suggest the authors modify it as:

A Machine learning-based Correlation Analysis Between Driver Behaviour and Vital Signs: Case study

  1. The introduction section should include several reference citations, especially in the first paragraph.
  2. The last paragraph of the introduction section includes many details; therefore, I suggest the authors summarize the content into several lines.
  3. In Table 1, it is better to mention the authors’ names before the citation, which would make the manuscript more understandable, such as [13]-> Villanueva et al [13].
  4. In line 236, it is better to put the variables’ descriptions together, such as:
  5. TP is True positive, TN donates False negative, ….. 
  6. Please check other equations.
  7. In line 398, From the table -> From table 3, we can find that ……
  8. English writing should be improved: 
  9. In line 398, the results look good for almost…-> the accuracy for all external factors except xxx exceeds 82% ….
  10. I suggest adding related citations after each method in Table 4.
  11. What SCV is? It is better to mention its full description (name) like other mentioned methods. 
  12. The manuscript mentioned that the proposed methodology adopted LSTM and CNN, however, the content has not provided details about this. Please modify Figure 2 to clearly describe how to use LSTM and CNN in this study.
  13. The whole manuscript needs to be carefully checked looking for English typos and grammar and modified accordingly; 
  • The whole manuscript needs to be carefully checked looking for English typos and grammar and modified accordingly; 

Author Response

Dear Reviewer,

Thank you for valuable comments. Please, find attached detailed answers to you comments.

Alexey

Reviewer 2 Report (Previous Reviewer 2)

The paper has well written 

Author Response

Dear Reviewer,

Thank you for positive estimation of our paper.

Alexey

Round 2

Reviewer 1 Report (Previous Reviewer 1)

Thanks to the authors for modifying the manuscript.

This manuscript is a resubmission of an earlier submission. The following is a list of the peer review reports and author responses from that submission.

Round 1

Reviewer 1 Report

The manuscript presents a study for Correlation Analysis Between Driver Vital Signs and Driver Behaviour. The idea is perfect, but the manuscript can be improved by considering the following comments.

1.   The title doesn’t include the methodology used in this study. To make the article‘s content being clearer, I suggest adding deep-learning words or any specific methods.

2.   The title should change Vehicle Behaviour -> Driver Behaviour.

3.   The abstract doesn’t mention the methodology of the algorithms used in the manuscript. Line 52 mentioned the authors used CNN and LSTM for data training, but nothing was introduced in the abstract. Therefore, I suggest the authors rewrite the abstract carefully, considering the background, gap, methodology, results, and value of the findings to the research area.

4.   No keywords found in the manuscript.

5.   There are many English grammar errors and academic writing errors, such as

In line 1, the paper present -> presents

Line 6, detect->detecting

In academic writing, try to not use personal pronouns and use passive sentences instead.

6.   In line 3, we developed new models….-> new models are developed….

7.   In line 5, We used our captured data set…. -> The captured dataset is used to ….

 the sensor data-> sensor data, all the variables->all variables, the data suggest-> the findings suggest.

8.   In the introduction section, the research gap of the study hasn’t been described clearly and there is a need to list the main contributions as well.

9.   in line 43, “subsection 3.1” should not be written in the introduction.

10.    In the first paragraph in the introduction section, “It is” is written many times, please rewrite the paragraph again in a better way.

11.    in line 122, Approach-> The proposed approach

12.    in line 123, This chapter -> This section

13.    in line 124, which lead, which leads

14.    in line 133, each frame, which frame, please make it clear.

15.    What is 3DDFA_V2?  mentioned in line 137?

16.    I suggest adding equations to the methodology in section 3, to make the content more understandable

17.    What do the authors mean by “(ax, ay, az, gx, gy, gz)” in line 324?

18.    In line 363, By analyzing Figure 6 -> By analyzing Figure 6

19.    we concluded that the model can-> we concluded that the proposed model can

20.    What are the main features used for this study and how can be extracted? And how did the authors perform data preprocessing before using the datasets for training the proposed model? Please mention this information in the manuscript.

21.    Figures 11-13 are in a bigger size; I suggest the authors to reduce their size.

22.    In line 488, insides ->insights

23.    The sentence in lines 488-490 is not understandable, please rewrite it again.

24.    The content of the conclusions section is written simply. Please rewrite it considering a description of the methodology, the significance of the study and results, and future work as well.

25.    English grammar errors and typos should be checked and modified; the whole manuscript should be checked carefully.

mentioned in the section of the comments and suggestions for authors 

Author Response

1

Reviewer 2 Report

This paper covered interesting topics, however, I have some suggestions for further improvement.

1. Motivation for this work needs to be included as separate section.

2. The introduction section needs improvement, added some additional information for new readers.

3. In related work, provides a summary table.

4.  Results need to be explained in convincing way.

5. It would be better if authors included further research directions.

Reviewer 3 Report

No comments - accepted in error.

Needs major work - sorry!